# ELSE: An Extremely Lightweight Adapter of Simple Expression for Vision Transformer Fine Tuning

## Abstract

Inspired by Parameter-Efficient fine tuning (PEFT) methods in natural language processing, numerous efforts have been made in seeking lightweight plug-in modules to adapt Vision Transformer (ViT) to downstream applications. However, the majority of such endeavor is motivated from the architecture design point of view, while neglects the training dynamics of fine tune in terms of efficiency and stability. In contrast, this study aims to investigate how fine tune affects architecture design, and turns out a lightweight module by theoretical and experimental parsing of fine tune. As observed by us, the parameter initialization of fine tune has a significant influence on the training stability and efficiency. We notice that initializing all the parameters of Adapter to zeros can make the fine tuning start approximately from the original ViT with desired training dynamics, due to the universal representation of ViT learnt from big data. Our theoretical deduction further shows that such initialization will cause gradient vanishing of Adapter, making the large portion of it inactive, which gives rise to the opportunity of simplifying the Adapter into an extremely lightweight equal form, that is, a single learnable vector instead of the full Adapter. To this end, an Extremely Lightweight adapter of Simple Expression (ELSE) can be approached. In the experiments, ELSE achieves superior or comparable transfer learning performance with less than $0.07\%$ of the model's parameters fine tuned while retaining the plug-and-play flexibility.

## 1 Introduction

With the remarkable success of Vision Transformers (ViTs), fine tuning pre-trained visual models for downstream tasks has become main-stream practice (Zhai et al., 2019; Xie et al., 2021; Chen et al., 2022). However, as the number of model parameters grows (Dehghani et al., 2023; Liu et al., 2022), the training cost for direct fine tuning becomes prohibitively high, and the need to leverage a full set of weights for each task separately causes significant complexity for large-scale model transfer. Linear probing and bias-based model training (Zaken et al., 2021) often yield suboptimal results because downstream task data are rarely compatible with the model's internal parameters or output, say, domain shift to pre-training.

Inspired by the extensive researches on fine tuning pre-trained models in natural language processing (NLP) (Devlin et al., 2019; Brown et al., 2020), visual researchers have focused on proposing various Parameter-Efficient fine tuning (PEFT) methods, which address the issue of reducing the number of adjustable model parameters in transfer learning. These methods involve either inserting additional trainable modules, such as adapters (Chen et al., 2022) or prompt tokens (Jia et al., 2022) into the frozen ViT (Dosovitskiy et al., 2021) backbone, or altering the weight matrix of the pre-trained model (such as LoRA (Hu et al., 2021)). Subsequent researches struggle to either refine these methods (Zhao et al., 2024; Jie & Deng, 2023; Jiang et al., 2024) or broaden their applicability (Yin et al., 2023; Xin et al., 2024; Wang et al., 2024a; Marjit et al., 2024).

The emergence of AdaptFormer (Chen et al., 2022) brought in an innovation to the literature due to its overall excellent performance. However, a new problem arises from its training: Prior to fine tune, as claimed, the module must follow a strict restriction to initialize the parameters of the newly introduced module by imposing partially zero setting not far from identify function. Otherwise, the

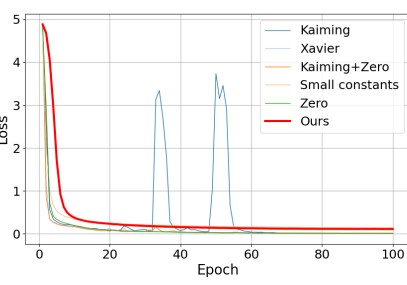

(a) Loss stability on CIFAR-100 dataset

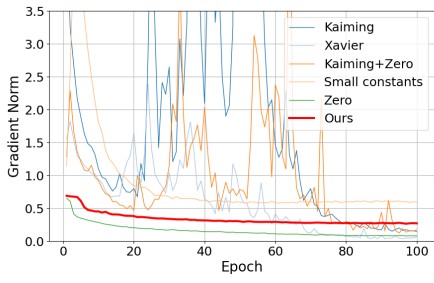

(b) Gradient update stability on CIFAR-100 dataset

Figure 1: Loss-indicated stability and gradient update-implied stability of Adapter with different initialization matrices on the CIFAR-100 dataset. The bold red curve represents our proposed method, which aligns with the Adapter initialized with all zeros and maintains stability during training.

training stability will not be certainly ensured. In fact, except for light weight and precision, training stability as well as time consuming for adaptation is also a key factor to affect user's preference in choosing the existing methods for rendering their solutions towards downstream deployments. So, this investigation stems from studying how initialization of model parameters affects training stability, which turns out the proposed new Adapter termed ELSE aiming to promise light weight, precision, training stability, and efficiency as a whole. Figure 1 and Table 1 illustrate the time-varying loss and gradient, as well as the final classification performance when applying different metrics to initialize the Adapter. Here, we can see that zero initialization leads to stable gradient descend while ensures rapid loss decreasing. Meanwhile, Rezero (Bachlechner et al., 2021) shows that simply applying a zero-initialized gating on each residual connection to ensure initial dynamical isometry outperforms more complex approaches. This appeals our interest in studying zero setting of parameter initialization, because zero initialization also means starting adaptation from the original backbone of ViT to leverage its universal representations turned out from big data based training, providing a solid base for adaptation to various downstream scenarios. In such a circumstance, the training stability assured by starting the fine tuning from a controllable checkpoint is essential, and zero initialization fits into such demand to the best extent since it makes the plug-in module totally transparent to the full model at the very beginning. Yet, the Adapter structure faces a significant gradient vanishing problem with zero initialization. In this study, we reveal the cause of such vanishing gradient. This leads to an extremely lightweight module that can replace Adapter with a learnable vector only, based on the theoretical deduction from a training perspective. The experiments show that it promises superior or compatible performance with much fewer parameters to be fine tuned for downstream tasks.

Our main contributions are as follows: (1) We investigated the missing issue of how parameter initialization affects the stability of model fine tuning in PEFT, and reveal the importance of all-zero initialization for this sake. (2) We identified the cause of gradient vanishing in the Adapter's all-zero initialization, which leads to an extremely lightweight Adapter of simple expression (a single vector), namely ELSE, as a replacement of traditional Adapters, preserving its plug-and-play flexibility while achieving promising transfer learning performance with less than $0.07\%$ of the model's parameters fine tuned for downstream tasks. (3) ELSE outperforms the other PEFT methods in an overall sense if considering classification precision against parameter number, and promises better few-shot learning and domain generalization in some scenarios.

Table 1: Performance of Adapter with various initialization methods.

| Initial mode | Image Accuracy (%) ↑ | | |
|---|---|---|---|
| | CIFAR-100 | SVHN | Food-101 |
| **Kaiming** | 89.96 | 96.50 | 86.88 |
| **Xavier** | 91.04 | 96.30 | 86.93 |
| **Xavier + Zero** | 91.17 | 96.76 | 86.95 |
| **Small constants** | 91.36 | **97.22** | 86.91 |
| **Zero** | **91.67** | 94.90 | **87.87** |

## 2 RELATED WORK

### 2.1 VISION TRANSFORMER

Transformer (Vaswani et al., 2017) was first introduced in natural language processing, giving rise to the rapid development of pre-trained large-scale models (Brown et al., 2020; Gao et al., 2024). Motivated by its immense success, ViT (Dosovitskiy et al., 2021) was the first to introduce Transformer into computer vision. Transformer's ability to model long-range dependencies has significantly advanced its applications in computer vision, with appealing performance in a wide range of tasks, such as image classification (Dosovitskiy et al., 2021; Liu et al., 2021), object detection (Zeng et al., 2022; Chi et al., 2020), semantic segmentation (Jain et al., 2023; Xu et al., 2022), and video processing (Li et al., 2023; Ju et al., 2022). In addition, Transformers have enhanced the visual recognition capabilities through a large-scale pre-training (Chen et al., 2021; Ni et al., 2022; Wang et al., 2024b). So far, a crucial challenge is how to fine tune large visual models to adapt them to specific downstream tasks.

### 2.2 PARAMETER-EFFICIENT FINE TUNE

PEFT aims to fine tune a minimal set of parameters, to adapt large-scale pre-trained models to downstream target tasks. In the context of NLP (Devlin et al., 2019; Brown et al., 2020), researchers have explored several methods for parameter-efficient transfer learning. The pioneering works include the token-based method (Li & Liang, 2021; Lester et al., 2021) and the network-based method (Houlsby et al., 2019; Zaken et al., 2021). The former usually adds a certain number of prefix tokens to the tokens in the multi-head self-attention (MHSA) layers, using additional token representations to help the pre-trained model adapt to downstream tasks. The latter usually integrates shallow module, such as Adapter (Houlsby et al., 2019) into the pre-trained model, in order to adapt the overall representations to downstream tasks after fine tuning the plug-in modules.

With the emergence of large-scale visual data (Sun et al., 2017; Deng et al., 2009), PEFT has also received much attention in the visual literature. VPT (Jia et al., 2022) first introduced prompt tuning into ViT, which added a small number of learnable tokens to the input while freezing the rest. Subsequently, various prompt-base methods (Bahng et al., 2022; Nie et al., 2023; Zhu et al., 2023; Wang et al., 2024a; Tang et al., 2024) have been developed. However, the performance of prompt is severely hindered by the number of the tokens, and different tasks require different amounts of new tokens to adapt. LoRA (Hu et al., 2021) introduces the re-parametrization method (Jiang et al., 2022; Marjit et al., 2024; He et al., 2023b; Basu et al., 2024), which applies two trainable low-rank matrices as proxy of the parameter matrix so as to reduce the number of parameters for fine tuning. However, it may have limited performance when dealing with high-dimensional features. SSF (Lian et al., 2022) performs learnable translation and scaling on the features of Transformer after each operation, yielding a large number of additional operations. AdaptFormer (Chen et al., 2022) more formally introduces Adapter into ViT, leading to a series of Adapter-based methods (Jie & Deng, 2022; Pan et al., 2022; Yang et al., 2023; Mou et al., 2024; Zhao et al., 2024). The concept involves parallely combining a learnable down projection layer and an up projection layer in order alongside the MLP within the Transformer block, and then adding the scaled output to the MLP's output. However, both AdaptFormer (Chen et al., 2022) and its subsequent works directly introduce the Adapter into ViT, without studying the mechanism of its fine tuning. AdaptFormer also pointed out that the new function obtained by introducing initial values into the Adapter part cannot deviate too much from the original identity function. Otherwise, training instability may occur. After rethinking its initialization method's training stability by theoretical parsing, we propose to use a more lightweight component referred to as ELSE, which replaces the original complex structure with just one vector.

## 3 PRELIMINARIES

For classification tasks, a standard ViT comprises a patch embedding layer, multiple interconnected Transformer blocks, and a final classification head. The Transformer blocks are illustrated in Figure 2(a). Given an image $x \in \mathbb{R}^{H \times W \times 3}$, the patch embedding layer splits and flattens the sample $x$ into patches $x_p \in \mathbb{R}^{N \times (P^2 d)}$, where $W$ and $H$ represent the height and width of the input image, respectively, $P^2$ is the resolution of each patch, $d$ denotes the output channel of the embedding layer,

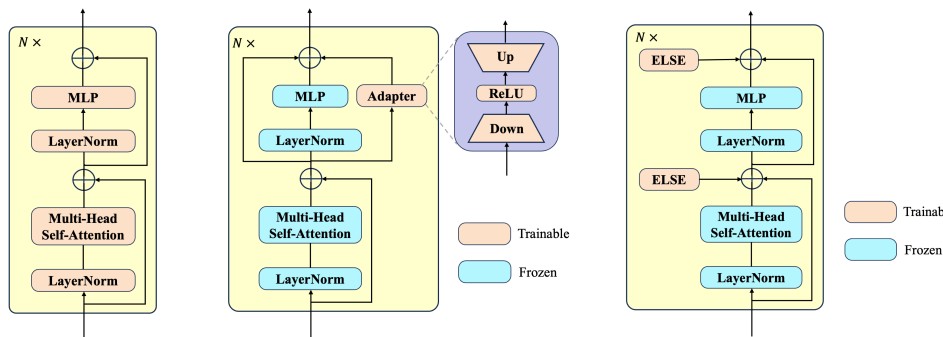

(a) Full fine tuning in ViT    (b) Adapter based fine tuning in ViT    (c) ELSE based fine tune in ViT

Figure 2: Adapter has added a parallel branch alongside the MLP in the Transformer block, containing a trainable bottleneck module. As shown in 2(c), we replace the original Adapter structure with a learnable vector $v \in \mathbb{R}^d$, namely ELSE, with the two plug-in locations depicted. During fine tuning, the blue part is frozen, while the orange part undergoes training. ELSE starts its training from zero initialization.

and $N = HW/P^2$ is the number of image tokens. The pre-added `[cls]` token and image tokens are mixed and fed to Transformer blocks for self-attention computation.

Each Transformer Block comprises a multi-head self-attention layer (MHSA) and an MLP layer. In MHSA, the tokens are linearly projected into three matrixs, namely $\mathbf{Q}$, $\mathbf{K}$, and $\mathbf{V}$. The process of self-attention computation is as follows:

$$x'_\ell = Attention(\mathbf{Q}, \mathbf{K}, \mathbf{V}) = Softmax(\frac{\mathbf{Q}\mathbf{K}^T}{\sqrt{d}}\mathbf{V}) \tag{1}$$

where $x'_\ell$ are the tokens produced by MHSA at the $\ell$-th layer, and $d$ is a scaling constant. Then, the output $x'_\ell$ will be forwarded to the subsequent LayerNorm and MLP layer, where the MLP layer comprises two fully connected layers with GELU activation function in between. This can be formulated as follows:

$$x_\ell = MLP(LN(x'_\ell)) + x'_\ell \tag{2}$$

where $x'_\ell$ is the output of the $\ell$-th Transformer block. The `[cls]` will be fed to the classifier for final visual recognition in the last layer.

To efficiently transfer the pre-trained model to downstream tasks, as illustrated in Figure 2(b), the Adapter introduces a branch alongside the MLP layer of the original Transformer block, while freezing the remaining parameters in the Transformer block. This add-in branch is a lightweight module tailored for task-specific fine tuning. It comprises a down projection layer $W_{down} \in \mathbb{R}^{d \times \hat{d}}$ and an up projection layer $W_{up} \in \mathbb{R}^{d \times \hat{d}}$, where $\hat{d}$ represents the intermediate dimension of the bottleneck, satisfying $\hat{d} \ll d$. Moreover, ReLU layer is employed between the two projection layers to introduce non-linearity. Upon the incorporation of the Adapter structure, the output of each Transformer block is formulated below:

$$x_\ell = MLP(LN(x'_\ell)) + Up(ReLU(Down(x'_\ell)) + x'_\ell \tag{3}$$

## 4   ELSE

AdaptFormer (Chen et al., 2022) indicates that if the initialization is too far from the identity function, the model cannot be trained stably. Therefore, they initialize the weights of the down projection layer using Kaiming Normal (He et al., 2015), and initialize all the other weights to zeros. This makes the full model based on such partially nonzero initialization of Adapter closely resemble the original ViT expressed function. In fact, it is essential to initialize the Adapter to all zeros to make it transparent to the full model at the very beginning such that the downstream adaptation can start

fine tuning from the original representations of ViT, which are universally promising due to the extensive pre-training on large-scale data. However, this often leads to the vanishing gradient problem as revealed below.

Without loss of generality, let's assume that it is a classification task. Here, $L$ represents the loss function, and $x$ and $x'$ denote the output of the final Transformer block and that generated by the final MHSA, respectively. We can then abbreviate Eq. (3) as:

$$x = x_{MLP} + x_{Adapter} + x' \tag{4}$$

where $x_{MLP}$ is the output of the MLP layer, and $x_{Adpter}$ is that of the Adapter. The forward propagation of the of Adatper part can be expressed as:

$$
\begin{aligned}
x_{\text{Down}} &= \text{ReLU}(W_{\text{Down}} \cdot x' + b_{\text{Down}}), \\
x_{\text{Adapter}} = x_{\text{Up}} &= W_{\text{Up}} \cdot x_{\text{Down}} + b_{\text{Up}}.
\end{aligned}
\tag{5}
$$

When considering initial back propagation, the gradient of the loss function with respect to the output of the Transformer block is $\frac{\partial L}{\partial x}$. According to the chain differentiation rule, $\frac{\partial L}{\partial x_{Up}} = \frac{\partial L}{\partial x}$, where the rest part of the output is omitted since frozen after pre-training. For the learnable parameters in Adatper, their gradients are as follows:

$$
\begin{aligned}
\frac{\partial L}{\partial W_{Up}} &= \frac{\partial L}{\partial x_{Up}} \cdot x_{ReLU}^{T}, \quad \frac{\partial L}{\partial b_{Up}} = \frac{\partial L}{\partial x_{Up}}, \\
\frac{\partial L}{\partial x_{ReLU}} &= W_{Up}^{T} \cdot \frac{\partial L}{\partial x_{Up}}, \quad \frac{\partial L}{\partial x_{Down}} = \frac{\partial L}{\partial x_{ReLU}} \cdot ReLU'(x_{Down}), \\
\frac{\partial L}{\partial W_{Down}} &= \frac{\partial L}{\partial x_{Down}} \cdot x'^{T}, \quad \frac{\partial L}{\partial b_{Down}} = \frac{\partial L}{\partial x_{Down}}
\end{aligned}
\tag{6}
$$

Since the parameters are initialized to all zeros, $x_{Down} = 0$ and $x_{ReLU} = 0$. Consequently, all gradients in Eq. (6) can be simplified:

$$
\begin{aligned}
\frac{\partial L}{\partial W_{\text{Up}}} &= 0, \quad \frac{\partial L}{\partial b_{\text{Up}}} = \frac{\partial L}{\partial x}, \quad \frac{\partial L}{\partial x_{\text{ReLU}}} = 0, \\
\frac{\partial L}{\partial x_{\text{Down}}} &= 0, \quad \frac{\partial L}{\partial W_{\text{Down}}} = 0, \quad \frac{\partial L}{\partial b_{\text{Down}}} = 0
\end{aligned}
\tag{7}
$$

This implies that during training, only the bias of the up projection layer in the Adapter is learnable, while all the other parameters, initially set to 0, will remain at 0.

Considering the efficiency of fine tuning parameters, consequently, we propose ELSE to replace the entire Adapter structure. As show in Figure 2(c), ELSE consists of a single trainable vector $v$, where $v \in \mathbb{R}^d$, precisely corresponding with the bias of the up projection layer in the aforementioned Adapter. We adopted the original Adapter concept to plug in an ELSE after each of the MHSA layer and the subsequent MLP layer in every Transformer block. Accordingly, Eq. (1) and Eq. (3) are reformulated as follows:

$$x'_{\ell} = Attention(\mathbf{Q}, \mathbf{K}, \mathbf{V}) + v_1 \tag{8}$$

$$x_{\ell} = MLP(LN(x'_{\ell})) + v_2 + x'_{\ell} \tag{9}$$

where $v_1$ and $v_2$ are the single trainable vectors.

During the fine tuning phase, we optimize only the newly added vectors while keeping all the other parameters frozen. Specifically, the original model (blue block in Figure 2(c)) loads weights from the pre-trained checkpoint and keeps its parameters frozen. The newly added vectors (orange blocks) are updated based on the data from specific domains under corresponding task losses. Even with much fewer fine tuning parameters than those of Adapter, such a single unified model as defined in Eq. (8) and Eq. (9) can still accommodate broad-spectrum downstream tasks. Unlike the approaches of learning the biases (Zaken et al., 2021; Cai et al., 2020) of ViT directly while freezing the remaining part, the plug-in biases introduced by ELSE allow more flexible transfer learning because ELSE is independent of ViT, making its original representation learnt from big data preserved. In another words, the biases introduced by ELSE matter domain shift only to bridge the downstream gap, not altering the original representation of ViT.

## 5 EXPERIMENTS

### 5.1 EXPERIMENTS SETTINGS

**Pre-trained backbone.** All our experiments were conducted using ViTB/16 (Dosovitskiy et al., 2021), which was pre-trained on ImageNet21K (Ridnik et al., 2021) and utilized fully supervised learning, unless otherwise noted.

**Implementation Details.** We conducted all the experiments using the PyTorch toolkit timm on a single NVIDIA 3090 GPU (24G) under Ubuntu 18.04.5 LTS. To evaluate the performance of ELSE in transfer learning, we conducted experiments in 4 aspects: For VTAB-1K, we adhered to the experimental configurations of VPT (Jia et al., 2022) and NOAH (Zhang et al., 2022), resizing the images to $224 \times 224$ without applying any data augmentation; For few-shot learning and domian generalization, we adhered to the configurations and evaluation criteria of NOAH (Zhang et al., 2022); For the dense training allowable CIFAR-100, SVHN, and Food-101 datasets, we utilized only the RandomResizedCrop data augmentation technique; Otherwise, we adhered to the common configurations as for VTAB-1K.

Table 2: In terms of classification precision and light weight on VTAB-1K, our method allows the fewest trainable parameters to achieve SOTA or top-tier precision in most cases. "Group mean" denotes the average accuracy across three groups. "Full tuning" indicates fine tuning all parameters. "Linear" means that all parameters within the backbone are kept frozen. "Param. (M)" indicates the number of trainable parameters. "Mem. (GB)" notes the maximum training memory. "PPP*" represents the performance-per-parameter corrected using Normalized Accuracy. Bold and underlined indicate the best and second best performance, respectively.

| | Param. (M) | Mem. (GB) | Natural | | | | | | | Specialized | | | | Structured | | | | | | | | Group Mean | PPP* |
|---|---|---|---|---|---|---|---|---|---|---|---|---|---|---|---|---|---|---|---|---|---|---|---|
| | | | CIFAR-100 | Caltech101 | DTD | Flowers102 | Pets | SVHN | Sun397 | Camelyon | EuroSAT | Resisc45 | Retinopathy | Clevr-Count | Clevr-Dist | DMLab | KITTI-Dist | dSpr-Loc | dSpr-Ori | sNORB-Azim | sNORB-Elev | | |
| *Traditional methods (Acc.)* | | | | | | | | | | | | | | | | | | | | | | | |
| Full tuning | 85.84 | 9.40 | 68.9 | 87.7 | 64.3 | 97.2 | 86.9 | 87.4 | 38.8 | 79.7 | 95.7 | 84.2 | 73.9 | 56.3 | 58.6 | 41.7 | 65.5 | 57.5 | 46.7 | 25.7 | 29.1 | 68.96 | - |
| Linear | **0.04** | 3.09 | 63.4 | 85.0 | 63.2 | 97.0 | 86.3 | 36.6 | 51.0 | 78.5 | 87.5 | 68.6 | 74.0 | 34.3 | 30.6 | 33.2 | 55.4 | 12.5 | 20.0 | 9.6 | 19.2 | 57.64 | - |
| *Parameter-efficient tuning methods (Acc.)* | | | | | | | | | | | | | | | | | | | | | | | |
| Adapter | 0.20 | 6.53 | 69.2 | 90.1 | 68.0 | 98.8 | 89.9 | 82.8 | 54.3 | 84.0 | 94.9 | 81.9 | 75.5 | 80.9 | 65.3 | 48.6 | 78.3 | 74.8 | 48.5 | 29.9 | 41.6 | 73.85 | 2.6 |
| BitFit | 0.14 | 6.65 | 72.8 | 87.0 | 59.2 | 97.5 | 85.3 | 59.9 | 51.4 | 78.7 | 91.6 | 72.9 | 69.8 | 61.5 | 55.6 | 32.4 | 55.9 | 66.6 | 40.0 | 15.7 | 25.1 | 65.21 | 2.7 |
| LoRA | 0.29 | 6.98 | 67.1 | 91.4 | 69.4 | 98.8 | 90.4 | 85.3 | 54.0 | 84.9 | 95.3 | 84.4 | 73.6 | 82.9 | **69.2** | 49.8 | 78.5 | 75.7 | 47.1 | 31.0 | 44.0 | 74.60 | 2.5 |
| VPT | 0.64 | 8.13 | 78.8 | 90.8 | 65.8 | 98.0 | 88.3 | 78.1 | 49.6 | 81.8 | 96.1 | 83.4 | 68.4 | 68.5 | 60.0 | 46.5 | 72.8 | 73.6 | 47.9 | 32.9 | 37.8 | 71.96 | 2.1 |
| SSF | 0.24 | 7.47 | 69.0 | 92.6 | **75.1** | 99.4 | 91.8 | 90.2 | 52.9 | **87.4** | 95.9 | **87.4** | 75.5 | 75.9 | 62.3 | **53.3** | 80.6 | 77.3 | 54.9 | 29.5 | 37.9 | 75.69 | 2.6 |
| NOAH | 0.43 | 7.27 | 69.6 | **92.7** | 70.2 | 99.1 | 90.4 | 86.1 | 53.7 | 84.4 | 95.4 | 83.9 | 75.8 | 82.8 | 68.9 | 49.9 | **81.7** | 81.8 | 48.3 | 32.8 | 44.2 | 75.48 | 2.3 |
| ConvPass | 0.37 | 7.51 | 72.3 | 91.2 | 72.2 | 99.2 | 90.9 | **91.3** | 54.9 | 84.2 | 96.1 | 85.3 | 75.6 | 82.3 | 67.9 | 51.3 | 80.0 | **85.9** | 53.1 | 36.4 | **44.4** | **76.56** | 2.4 |
| AdaptFormer | 0.20 | 6.55 | 70.8 | 91.2 | 70.5 | 99.1 | 90.9 | 86.6 | 54.8 | 83.0 | 95.8 | 84.4 | **76.3** | 81.9 | 64.3 | 49.3 | 80.3 | 76.3 | 45.7 | 31.7 | 41.1 | 74.75 | 2.6 |
| RepAdapter | 0.22 | 6.77 | 72.4 | 91.6 | 71.0 | 99.2 | 91.4 | 90.7 | 55.1 | 85.3 | 95.9 | 84.6 | 75.9 | 82.3 | 68.0 | 50.4 | 79.9 | 80.4 | 49.2 | **38.6** | 41.0 | 76.09 | 2.6 |
| SPT-Adapter | 0.38 | 7.12 | 72.9 | 93.2 | 72.5 | 99.3 | 91.4 | 88.8 | 55.8 | 86.2 | 96.1 | 85.5 | 75.5 | **83.0** | 68.0 | 51.9 | 81.2 | 82.4 | 51.9 | 31.7 | 41.2 | 76.41 | 2.4 |
| Res-Tuning | 0.55 | 8.95 | 75.2 | 92.7 | 71.9 | 99.3 | 91.9 | 86.7 | **58.5** | 86.7 | 95.6 | 85.0 | 74.6 | 80.2 | 63.6 | 50.6 | 80.2 | 85.4 | **55.7** | 31.9 | 42.0 | 76.32 | 2.2 |
| **ELSE(Ours)** | 0.06 | 6.34 | **81.3** | 92.5 | 73.0 | **99.4** | 91.9 | 87.1 | 57.4 | 85.8 | **96.1** | 85.2 | 75.7 | 73.9 | 63.3 | 48.3 | 81.6 | 73.8 | 54.6 | 29.6 | 31.8 | 75.35 | **3.1** |

### 5.2 PERFORMANCE COMPARISON ON IMAGE CLASSIFICATION

**Datasets.** To evaluate adaptability, we conducted experiments on the VTAB-1K (Zhai et al., 2019) benchmark, which is commonly used for most visual PEFT tasks. The training data for this benchmark is limited, only 1000 samples per task. According to Zhai et al. (2019), we used the provided training set to determine hyper-parameters by dividing them into 800-200 segments, and ultimately used the model trained on the complete training set for testing.

**Baselines.** To date, a couple of adaptation strategies have emerged in the field of PEFT, each relying on specific architecture with trainable parameter number ranging from $10^{-2}$ to $10^0$ M. Here, we chose the most representative methods as baselines, such as VPT (Jia et al., 2022), LoRA (Hu et al., 2021), and BitFit (Zaken et al., 2021). Moreover, since our method stems from Adapter, we also compared it to the Adapter's family, including the methods directly applying Adapter such as AdaptFormer (Chen et al., 2022), RepAdapter (Luo et al., 2023), and SPT-Adapter (He et al., 2023a) and the variants of Adapter such as NOAH (Zhang et al., 2022), ConvPass (Jie & Deng, 2022), and Res-Tuning (Jiang et al., 2024).

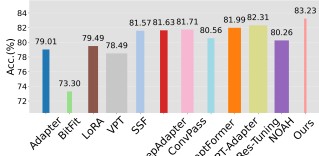
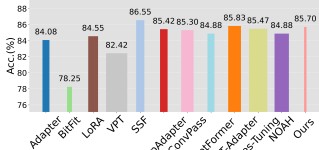
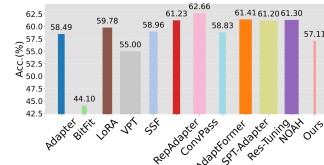

(a) Performance on the Natural group

(b) Performance on the Specialized group

(c) Performance on the Structured group

Figure 3: The average performance of the three groups in VTAB-1K is depicted by the bar chart, where the height represents accuracy and the width represents the number of model parameters.

**Results.** Table 2 presents our top-1 accuracy on VTAB-1K, along with a comparison of the light weight degree to the previous methods. We can observe that no method can achieve SOTA performance across all 19 tasks and there is a compromise between light weight and classification precision due to their contradicting requirements on parameter number. For example, although some methods like ConvPass and SPT-Adapter yield satisfactory classification results, their outcomes are obtained at the cost of significantly increasing the number of parameters. In response to this situation, we adhere to SiVA's proposal (Agrawal et al.) to employ *performance per parameter* (PPP) to measure the overall performance taking into account both parameter number and classification precision. To eliminate the randomness caused by the varying category number of different tasks in VTAB-1K, as done in ACC-NORM (Wu et al., 2024) and BayesTune (Kim & Hospedales, 2023), we utilize the *Normalized Accuracy* to alleviate the randomness, leading to

$$\text{PPP*} = log_{10}(\frac{1}{K} \cdot \frac{1}{N} \sum_{t=1}^{N} \frac{A_t - P_t^{\text{chance}}}{1 - P_t^{\text{chance}}}) \tag{10}$$

where $K$ represents the number of learnable parameters (unit: M), $P_t^{chance} = 1/C_t$, and $A_t$ and $C_t$ denote the accuracy and the number of categories for task $t$, respectively. As shown in Table 2, it is evident that our PPP* index is the highest, significantly surpassing the other methods, which means that our method can provide the best overall performance if taking into account light weight and classification precision simultaneously. In terms of maximum training memory usage, it is secondary only to linear probing and surpasses the other methods.

Figure 3 illustrates averaged classification precision against parameter number on the 3 scenarios listed in Table 2. It is straightforward that no method can promise classification and light weight simultaneously due to the contradicting requirements on parameter number. We can observed that the performance of ELSE varies over the 3 groups. For the Natural and Specialized groups, it achieves the best classification with only 10% parameter number against that of Res-Tuning (Jiang et al., 2024), and the second best classification with only 25% parameter number against that of SSF (Lian et al., 2022), respectively. ELSE's performance on the Structured group is less satisfactory, but still outperforms complete fine tuning, BitFit (Zaken et al., 2021), and VPT (Jia et al., 2022). We attribute this to the extremely fewer parameters of ELSE, which limits its learning capability. Even though, for the datasets other than dSpr-Loc and sNORB-Elev in this group, ELSE is not significantly inferior to the other solutions.

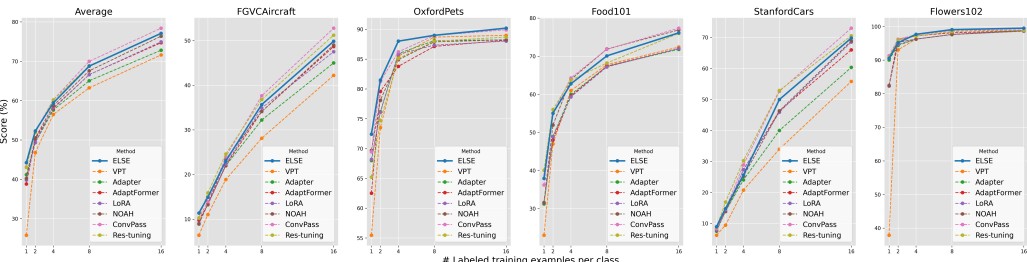

Figure 4: Results of few-shot learning on five fine-grained visual recognition datasets.

### 5.3 PERFORMANCE COMPARISON ON FEW-SHOT LEARNING

**Datasets.** To assess the our method in few-shot classification, we conducted experiments on 5 commonly used fine-grained datasets, including FGVC-Aircraft (Maji et al., 2013), Food-101 (Bossard et al., 2014), Oxford Flowers (Nilsback & Zisserman, 2006), Oxford Pets (Parkhi et al., 2012), and Stanford Cars (Krause et al., 2013). According to (Zhang et al., 2022), we conducted experiments with shot settings of 1, 2, 4, 8, and 16. The results are the averages of 3 trials conducted with different seeds.

**Results.** As illustrated in Figure 4, ELSE achieves the best performance across all settings on the Flowers102 and OxfordPets datasets. Furthermore, ELSE's superiority becomes more protruded when there are fewer training samples. For instance, when the shot is set to 1 or 2, ELSE outperforms all the previous methods. The advance of ELSE degrades with the shot number but it still ranks in top tier. We attribute this to the extremely small parameter number in ELSE, as tuning a smaller set of parameters needs in general fewer data.

### 5.4 PERFORMANCE COMPARISON ON DOMAIN GENERALIZATION

**Datasets.** Given the ubiquitous requirement of domain transfer, we adhered to the previous research (Zhang et al., 2022) and trained the model on ImageNet (Deng et al., 2009), utilizing 16 shots per category. Then, we directly evaluated it on 4 variants of ImageNet: (i) ImageNetv2 (Recht et al., 2019), which originates from a source different from ImageNet but adheres to the same collection protocol, (ii) ImageNet-Sketch (Wang et al., 2019), composed of sketch images for the same 1000 classes from ImageNet, (iii) ImageNet-A (Hendrycks et al., 2021), featuring images filtered through adversarial methods, and (iv) ImageNet-R (Hendrycks et al., 2020), a reproduction of ImageNet. ImageNet-A and ImageNet-R contain 200 categories, which are derived from the 1000 categories in ImageNet.

Table 3: The result of domain generalization: ELSE still exhibits competitive generalization capabilities on 4 domain transfer datasets.

| | Source | Target | | | | |
|---|---|---|---|---|---|---|
| | ImageNet | -V2 | -Sketch | -A | -R | mean |
| Adapter | 70.5 | 59.1 | 16.4 | 5.5 | 22.1 | 25.8 |
| VPT | 70.5 | 58.0 | 18.3 | 4.6 | 23.2 | 26.0 |
| LoRA | 70.8 | 59.3 | 20.0 | 6.9 | 23.3 | 27.4 |
| NOAH | 71.5 | 66.1 | 24.8 | **11.9** | 28.5 | 32.8 |
| Res-Tuning | 77.3 | 65.2 | **27.4** | 10.7 | 26.5 | 32.4 |
| **ELSE** | **77.3** | **66.4** | 25.9 | 11.7 | **28.7** | **33.2** |

**Results.** As shown in Table 3, we compared ELSE with several baselines common in domain adaptation. Compared to these methods, our results on ImageNet were unexpectedly high, achieving a 5.8% improvement over NOAH (Zhang et al., 2022), and thus demonstrating superior robustness to domain shift. When compared to the latest methods (Jiang et al., 2024), we achieved comparable or even better performance. This demonstrates ELSE's advantage over the existing methods in terms of cross-domain robustness.

### 5.5 PERFORMANCE ON COMPLETE DATASET & TRAINING STABILITY

**Datasets.** To evaluate the transfer performance of ELSE under sufficient training data conditions, as well as its overall training stability, we conducted experiments on 3 comprehensive and commonly used image classification datasets, namely, CIFAR-100 (Krizhevsky, 2009), SVHN (Goodfellow et al., 2013), and Food-101 (Bossard et al., 2014).

Table 4: Top-1 accuracy of ELSE on different complete datasets

| Method | Param.(M)↓ | Image Accuracy (%) ↑ | | |
|---|---|---|---|---|
| | | CIFAR-100 | SVHN | Food-101 |
| **Full fine tune** | 85.85 (100%) | 89.12 | 95.41 | **90.96** |
| **Linear** | **0.05** (0.06%) | 85.96 (-3.16) | 55.36 (-40.05) | 88.14 (-2.82) |
| **AdaptFormer** | 1.26 (1.47%) | 91.86 (+2.74) | **97.29** (+1.88) | 90.89 (-0.07) |
| **VPT** | 0.69 (0.80%) | 90.97 (+1.85) | 92.77 (-2.64) | 90.16 (-0.80) |
| **ELSE** | 0.07 (0.08%) | **92.26** (+3.14) | 95.67 (+0.26) | 88.60 (-2.36) |

Table 5: The average results on VTAB-1K test when ELSE was applied to various positions within the transformer block.

| Insertion position | Layer | Params. (M) ↓ | Group mean ↑ |
|---|---|---|---|
| **mid** | 1→12 | **0.01** | 74.03 |
| **tail** | 1→12 | 0.01 | 73.85 |
| **both** | 1→12 | 0.02 | **75.35** |
| **both** | 1→6 | 0.01 | 73.92 |
| **both** | 7→12 | 0.01 | 73.11 |

Table 6: We also examined the performance against various number of ELSEs, where $n$ represents the number of tokens produced by MHSA, $n + 1$ means including [cls] token. We report their average performance across the VTAB-1K benchmark.

| # ELSE | Params. (M) ↓ | Group mean ↑ |
|---|---|---|
| **1** | **0.02** | **75.35** |
| **n** | 3.61 | 73.71 |
| **n+1** | 3.62 | 74.26 |

**Results.** As shown in Table 4, compared to merely linear probing, we increased the trainable parameters up to only 0.08%, but the effect is remarkable in promising much better classification accuracy. This demonstrates that, even with a limited set of learnable parameters, our solution can promise the overall adaptation based on the downstream data, rather than solely relying on the classification head. Similar to the other PEFT methods, our approach can achieve comparable performance to that achieved with full fine tuning on complete datasets, and even surpass it on CIFAR-100.

Hereafter, we evaluate the training stability using the average loss caused by the trainable parameters during fine tuning as well as the variation of the L2 norm of the the gradients of such trainable parameters. As illustrated in Figure 1, aligned with the zero initialization of Adapter, ELSE also exhibits the same training stability.

## 5.6 ABLATION STUDIES

We examined the impact of ELSE's plug-in positions. For each Transformer block, there are 3 insertion options: "mid", "tail", and "both", corresponding to insertion after either MHSA or MLP, or both. As shown in the Table 5, inserting ELSE into every layer yields the best performance, and applying ELSE to both positions is not trivial.

As demonstrated in Table 6, optimal performance is attained when ELSE applies a uniform offset to all tokens, rather than assigning a unique offset to each one. We attribute this to the following consideration: The performance gap between pre-training and downstream scenario is in general caused by domain shift and the offset-based compensation introduced by ELSE can bridge such domain shift to some extent. It introduces an incremental learning paradigm by applying simple plug-in bias and zero initialization in addition to the pre-trained ViT without interference to its internal structure and representations, which makes the fine tuning more stable.

## 5.7 VISUAL ANALYSIS

We conducted attention visualization on the SVHN (Goodfellow et al., 2013) and CIFAR100 (Krizhevsky, 2009) datasets. As illustrated in the figure 5, it is evident that the original pre-trained network (such as ViT) has limited adaptability to downstream tasks. Its attention struggles to focus on semantic regions more relevant to downstream image representation. The incorporation of ELSE bridges the gap between the semantics of the upstream pre-trained model and downstream tasks, enabling us to adapt well to downstream tasks while training only a minimal number of parameters. For more visual analysis, please refer to the appendix.

## 6 CONCLUSION

Our research starts from investigating the training dynamics of Adapter in PEFT context. We identified the cause of gradient vanishing in Adaptper when initialized to all zeros, and subsequently introduced the more compact ELSE, solely composed of a single vector as a replacement. This new vector maintains overall excellent performance throughout the fine tuning process, which is an ex-

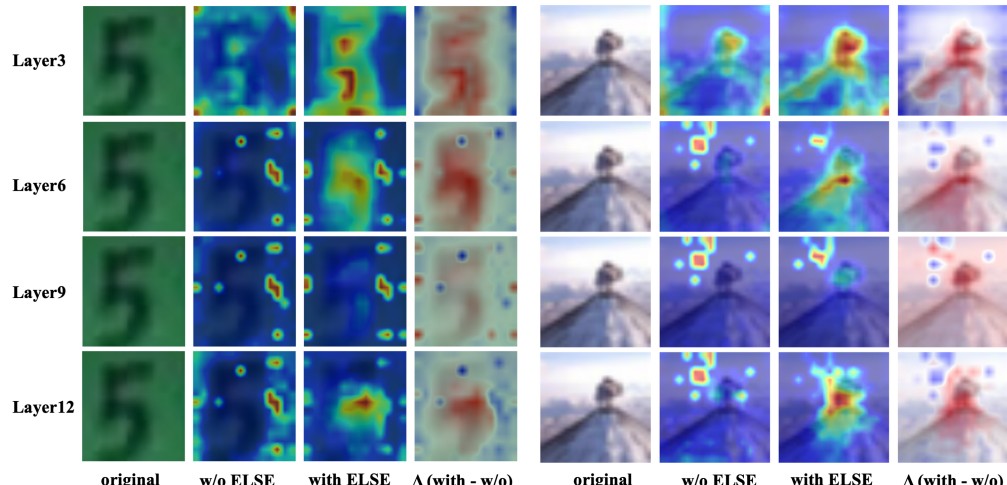

Figure 5: We selected the typical layers 3, 6, 9, and 12 from ViT and visualized their attention mechanisms. The 16 small images on the left represent the results on the SVHN dataset, while the 16 small images on the right represent the results on the CIFAR100 dataset. In the images for each dataset, the first column shows the original image, the second column shows the attention map without ELSE, the third column shows the attention map with ELSE, and the fourth column shows the difference between the third and second columns.

tremely lightweight plug-in module as far as we know. Since we propose all-zero initialization of ELSE, it is thus totally transparent to the full model at the very beginning of fine tune, bringing in the benefit to make the adaptation start from the original representations of pre-trained ViT, which is in general universally promising due to its extensive pre-training on big data.

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

## A APPENDIX

### A.1 SUPPLEMENTARY VISUALIZATION

To further understand the working mechanism of ELSE across different hierarchical spaces, we systematically analyzed the L2 norms of two learnable vectors (ELSE1 and ELSE2) at each layer. The results, as shown in the figure 6, reveal a highly consistent yet task-dependent distribution characteristic of ELSE across different tasks: In tasks with low semantic complexity and dominated by local structures (such as SVHN), the norms of ELSE are primarily concentrated in the shallow layers, indicating that the model relies more on the adjustment of low-level local features; in tasks with medium semantic complexity (such as CIFAR-100), ELSE maintains a certain magnitude of offset in both shallow and deep layers, reflecting the common adaptation requirements of tasks to multi-level feature spaces; and in tasks with high semantic complexity and strong diversity (such as Food-101), the norms of ELSE significantly increase in the deep layers, indicating that the model requires strong task-specific alignment in the semantic decision space. More importantly, the distribution patterns of ELSE1 and ELSE2 are not identical: ELSE1 (inserted after self-attention) is more biased towards the adjustment of structural representations and attention distributions, while ELSE2 (placed after MLP) primarily undertakes the function of redirecting semantic spaces. The two form a complementary hierarchical adjustment structure. The above results suggest that ELSE

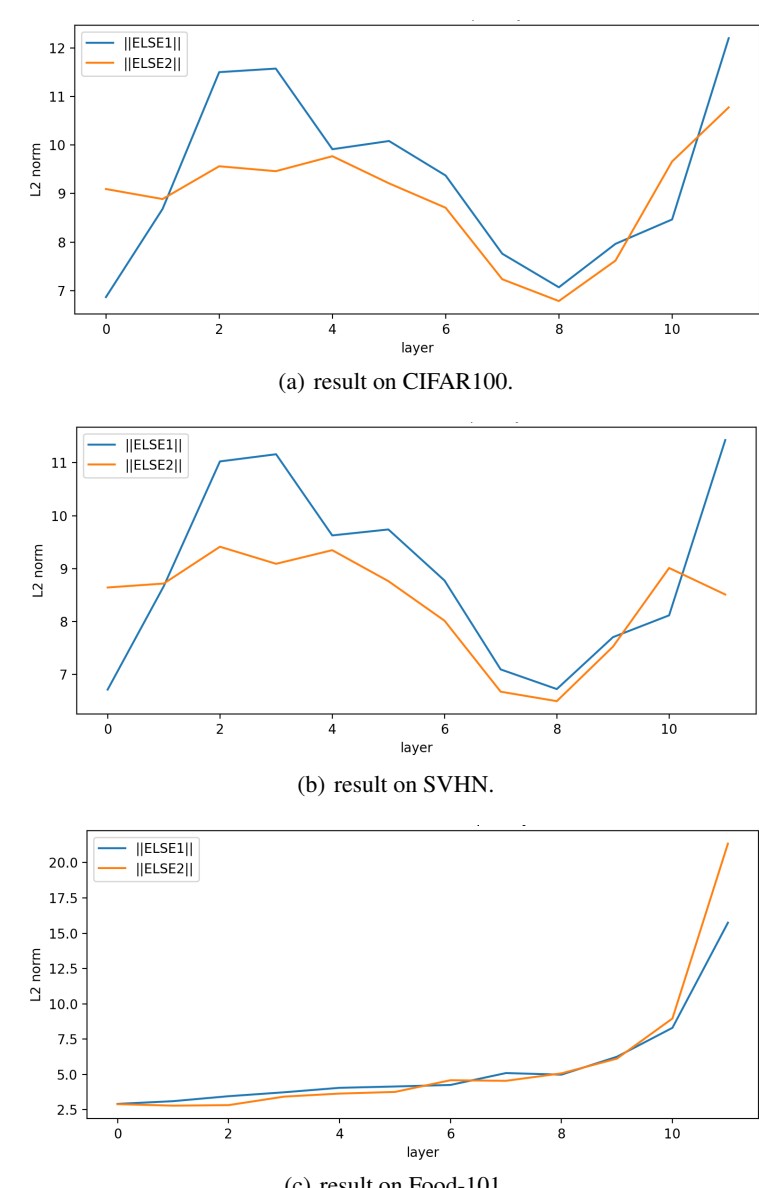

(a) result on CIFAR100.

(b) result on SVHN.

(c) result on Food-101.

Figure 6: L2 norm statistics of two ELSEs in different layers of ViT.

does not simply impose additive perturbations on ViT features, but learns a set of semantic offset directions with clear functional divisions at each layer, which can adaptively distribute according to task requirements, thereby effectively reshaping the frozen model. This interpretable hierarchical feature adjustment pattern differs from the difficult-to-observe matrix-level parameter perturbations in existing PEFT methods (such as BitFit, SSF, or LoRA).

In further analyzing the impact of ELSE on the model's decision space, we tested its response to the target class logit under different scaling coefficients ($\alpha$). As shown in the figure 7, the logit exhibits a consistent and approximately monotonic trend as $\alpha$ varies, indicating that increasing the offset along the direction learned by ELSE significantly enhances the logit for that class, while scaling in the opposite direction reduces the model's confidence in that class. This phenomenon suggests that ELSE does not simply apply a simple additive bias in the feature space, but rather learns a clear, task-specific semantic direction. The model can adjust the position of the classification boundary by linearly controlling in this direction. Furthermore, there are significant differences in the sensitivity of ELSE to different tasks: the slope of logit to $\alpha$ is highest on Food-101, indicating that this task

relies on deep semantic shifts; CIFAR100 has the second highest sensitivity; and SVHN shows the weakest change, reflecting its primary reliance on shallow structural features. This is highly consistent with the aforementioned L2 norm analysis, collectively demonstrating that ELSE exhibits a stable, reasonable, and interpretable behavior pattern when adapting to tasks at different semantic levels.

## A.2 DATASETS

Refer to Table 7. During the hyperparameter tuning, the 1000 training samples of VTAB-1K are further divided into a training set (800 samples) and a validation set (200 samples). The final result is generated by training on all 1000 training samples and evaluating on the test set.

Table 7: Statistics of used datasets.

| Dataset | # Classes | Train | Val | Test |
|---|---|---|---|---|
| **VTAB-1K** (Zhai et al., 2019) | | | | |
| CIFAR100 (Krizhevsky, 2009) | 100 | | | 10,000 |
| Caltech101 (Fei-Fei et al., 2004) | 102 | | | 6,084 |
| DTD (Cimpoi et al., 2014) | 47 | | | 1,880 |
| **Natural**   Oxford-Flowers102 | | 800/1,000 | 200 | |
| (Nilsback & Zisserman, 2006) | 102 | | | 6,149 |
| Oxford-Pets | | | | |
| (Parkhi et al., 2012) | 37 | | | 3,669 |
| SVHN (Goodfellow et al., 2013) | 10 | | | 26,032 |
| Sun397 (Xiao et al., 2010) | 397 | | | 21,750 |
| Patch Camelyon | | | | |
| (Veeling et al., 2018) | 2 | | | 32,768 |
| EuroSAT (Helber et al., 2019) | 10 | 800/1,000 | 200 | 5,400 |
| **Specialized**   Resisc45 (Cheng et al., 2017) | 45 | | | 6,300 |
| Retinopathy (Graham, 2015) | 5 | | | 42,670 |
| Clevr/count (Johnson et al., 2017) | 8 | | | 15,000 |
| Clevr/distance (Johnson et al., 2017) | 6 | | | 15,000 |
| DMLab (Beattie et al., 2016) | 6 | | | 22,735 |
| KITTI-Dist (Geiger et al., 2013) | 4 | 800/1,000 | 200 | 711 |
| **Structured**   dSprites/location | | | | |
| (Higgins et al., 2017) | 16 | | | 73,728 |
| dSprites/orientation | | | | |
| (Higgins et al., 2017) | 16 | | | 73,728 |
| SmallNORB/azimuth | | | | |
| (LeCun et al., 2004) | 18 | | | 12,150 |
| SmallNORB/elevation | | | | |
| (LeCun et al., 2004) | 18 | | | 12,150 |
| **Few-shot datasets** | | | | |
| Food-101 (Bossard et al., 2014) | 101 | | 20,200 | 30,300 |
| FGVC-Aircraft (Maji et al., 2013) | 100 | 1/2/4/8/16 | 3,333 | 3,333 |
| Oxford-Flowers102 | | per class | | |
| (Nilsback & Zisserman, 2006) | 102 | | 1,633 | 2,463 |
| Oxford-Pets | | | | |
| (Parkhi et al., 2012) | 37 | | 736 | 3,669 |
| Stanford Cars (Krause et al., 2013) | 196 | | 1,635 | 8,041 |
| **Domain generalization datasets** | | | | |
| ImageNet-1K (Deng et al., 2009) | 1,000 | 16 per class | - | 50,000 |
| ImageNet-V2 (Recht et al., 2019) | 1,000 | - | - | 10,000 |
| ImageNet-Sketch (Wang et al., 2019) | 1,000 | - | - | 50,889 |
| ImageNet-A (Hendrycks et al., 2021) | 200 | - | - | 7,500 |
| ImageNet-R (Hendrycks et al., 2020) | 200 | - | - | 30,000 |

## A.3 EXPERIMENTAL DETAILS

### A.3.1 PRE-TRAINED BACKBONE

Consistent with the prior PEFT approaches, we employ ViT-B/16 (Dosovitskiy et al., 2021) as the backbone, utilize the CLS token derived from the final layer for classification, and apply a classification head dedicated to each classification task.

### A.3.2 CODE IMPLEMENTATION

We use PyTorch and timm to implement all experiments on NVIDIA RTX 3090 GPUs under Ubuntu 18.04.5 LTS. Please refer to the supplementary material file for a detailed code.

### A.3.3 DATA AUGMENTATION

For VTAB-1K, we adhered to the experimental configurations of VPT (Jia et al., 2022) and NOAH (Hu et al., 2021), resizing the images to $224 \times 224$ without applying any data augmentation. For few-shot learning, following (Hu et al., 2021), for training samples, we use color-jitter and RandAugmentation; for validation/test samples, we resize them to $256 \times 256$, crop them to $224 \times 224$ at the center, and then normalize them with ImageNet's mean and standard deviation.

### A.3.4 HYPER-PARAMETERS

The experimental parameter settings for the discussion on Adapter initialization are shown in Table 8.

Table 8: Configuration of Adapter initialization for CIFAR-100, SVHN, and Food-101 datasets.

| Configuration | CIFAR-100 | SVHN | Food-101 |
|---|---|---|---|
| moddle dimension $\hat{d}$ | | 64 | |
| Optimizer | | AdamW (Loshchilov & Hutter, 2017) | |
| Base learning rate | | 1e-2 | |
| Weight decay | | 1e-4 | |
| Batch size | | 80 | |
| Training crop size | | 224 | |
| Learning rate schedule | | Cosine decay (Loshchilov & Hutter, 2016) | |
| GPU numbers | | 1 | |
| Warmup epochs | | 10 | |
| Training epochs | | 100 | |
| Augmentation | | RandomResizedCrop | |

The experimental parameter settings for ELSE in VTAB-1K are shown in Table 10. We adhere to the VPT (Jia et al., 2022) configuration, wherein no regularization is applied to the bias of trainable linear layers. Additionally, we have applied both L1 and L2 regularizations to all ELSE components, employing validation sets to identify the optimal hyperparameter configurations.

The experimental parameter settings for the complete datasets are shown in Table 9.

The experimental parameter settings for fewshot learning and domain generalization are shown in Table 11.

## A.4 ABOUT LARGE LANGUAGE MODELS

The translation, writing, and polishing of the paper were assisted by using an LLM (such as Chat-GPT).

Table 9: Configuration for complete CIFAR-100, SVHN, and Food-101 datasets.

| Configuration | CIFAR-100 | SVHN | Food-101 |
|---|---|---|---|
| Optimizer | AdamW (Loshchilov & Hutter, 2017) | | |
| Base learning rate | 1e-3 | | |
| Weight decay | 1e-2 | | |
| Batch size | 80 | | |
| Training crop size | 224 | | |
| Learning rate schedule | Cosine decay (Loshchilov & Hutter, 2016) | | |
| GPU numbers | 1 | | |
| Warmup epochs | 10 | | |
| Training epochs | 100 | | |
| Augmentation | RandomResizedCrop | | |

Table 10: configuration of ELSE for VTAB-1K banchmark.

| Configuration | VTAB-1K |
|---|---|
| Optimizer | AdamW (Loshchilov & Hutter, 2017) |
| Base learning rate | {0.01, 0.05, 0.1, 0.25, 0.5} |
| Weight decay of the whole | {1e-3,1e-4,1e-5} |
| L1 of ELSE | {1e-4, 1e-5, 0} |
| L2 of ELSE | {1e-3, 1e-4, 1e-5, 0} |
| Batch size | 64 |
| Training crop size | 224 |
| Learning rate schedule | Cosine decay (Loshchilov & Hutter, 2016) |
| GPU numbers | 1 |
| Warmup epochs | 10 |
| Training epochs | 100 |

Table 11: Configuration for few-shot and domain generalization.

| Configuration | Few-shot learning |
|---|---|
| Optimizer | AdamW (Loshchilov & Hutter, 2017) |
| Base learning rate | 5e-3 |
| Weight decay | 1e-5 |
| Batch size | 80 |
| Training crop size | 224 |
| Learning rate schedule | Cosine decay (Loshchilov & Hutter, 2016) |
| GPU numbers | 1 |
| Warmup epochs | 10 |
| Training epochs | 100 |
| Augmentation | RandomResizedCrop |

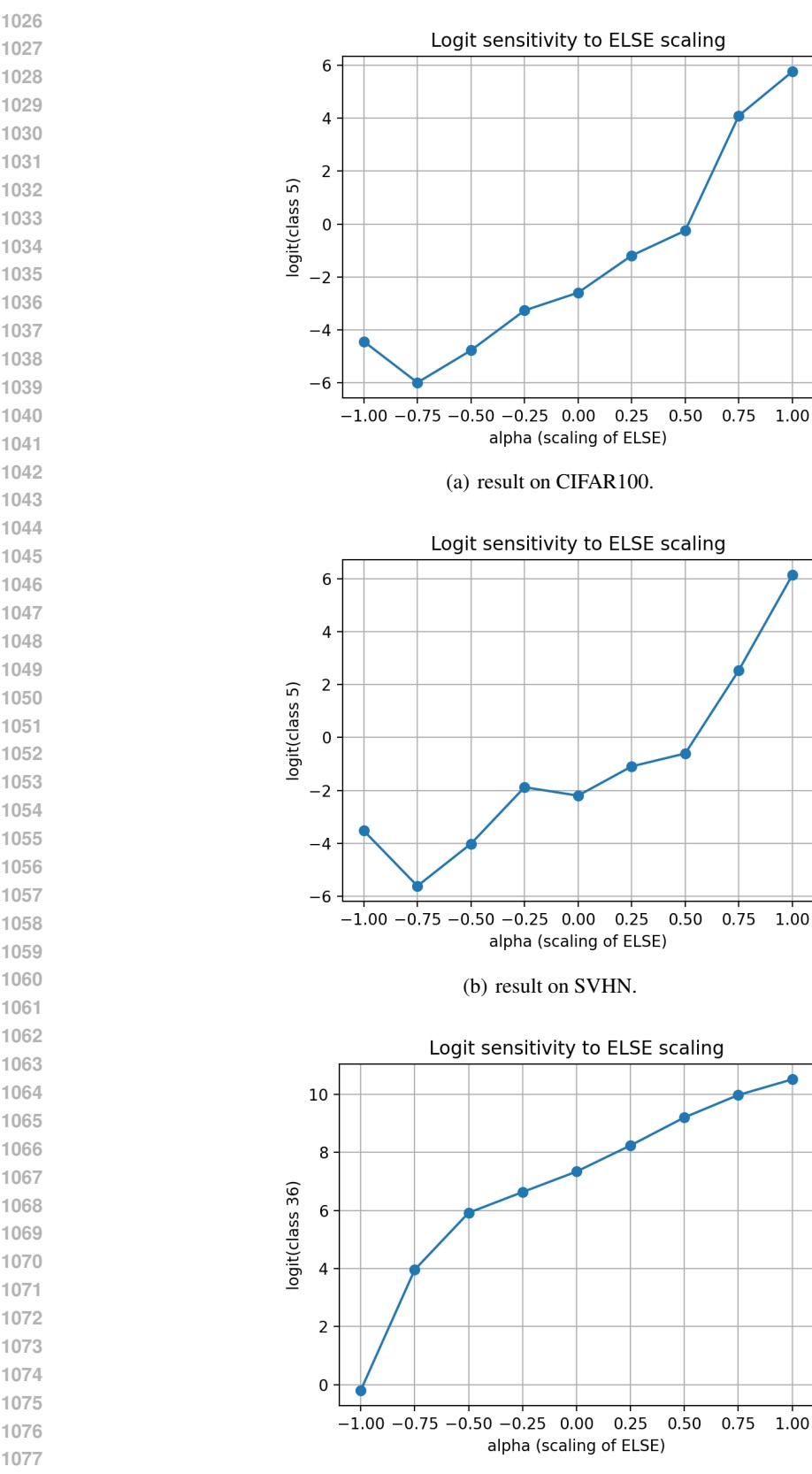

(a) result on CIFAR100.

(b) result on SVHN.

(c) result on Food-101.

Figure 7: The sensitivity of logit to ELSE scaling for a certain category in three datasets.

