# OpenReview forum: "ELSE: An Extremely Lightweight Adapter of Simple Expression for Vision Transformer Fine Tuning"
_ICLR.cc/2026/Conference — Submitted to ICLR 2026_

### Official Review · Reviewer_dizo · 2025-10-28

**Soundness:** 3
**Presentation:** 3
**Contribution:** 2
**Rating:** 4
**Confidence:** 5

**Summary:**

This paper proposes a new Parameter-Efficient Fine-Tuning (PEFT) method called ELSE (Enhanced Linear Shift Embedding). The approach introduces learnable additive vector (ELSE1 and ELSE2) after the attention and MLP layers in Vision Transformers, enabling efficient task adaptation with minimal additional parameters. The method is simple and compatible with pretrained ViTs such as MAE and MoCo. Experiments on VTAB-1K, CIFAR-100, SVHN, and Food-101 demonstrate consistent, though modest, performance gains.

**Strengths:**

1. The core idea of ELSE is extremely lightweight, introducing only two learnable additive vectors with almost no additional parameters or computational cost, making it highly suitable for large-scale vision tasks.
2. The paper covers a broad range of experiments, validating the method on multiple datasets and showing consistent performance improvements, which demonstrates the generality of the approach.
3. The paper is well-written and well-organized. The structure is logical, the language is clear, and the figures are easy to interpret. The appendix provides detailed experimental settings and hyperparameters, contributing to the overall high presentation quality.

**Weaknesses:**

1. The paper has limited novelty. The proposed additive bias concept is conceptually similar to existing methods such as Bias-Tuning, LoRA (low-rank approximation), and Adapter-based bias injection. The authors do not provide sufficient justification to distinguish ELSE from these approaches in a fundamental way.
2. The paper does not explain why the additive bias effectively facilitates task adaptation, nor does it provide feature visualizations or representational analyses to illustrate its underlying mechanism. The lack of theoretical or interpretability studies makes the method appear overly empirical.
3. Table 7 is too wide and exceeds the typesetting scope.

**Questions:**

The authors could further provide a theoretical explanation or mathematical analysis of the additive bias mechanism in the ELSE model, clarifying how it enhances feature representation and transferability. In addition, it is recommended to include visualization or attention map analyses to illustrate how the two learnable additive vectors behave under different inputs, thereby deepening the understanding of the model’s dynamic adjustment mechanism and interpretability.

---

> ### Author Response · Authors · 2025-11-19
> **Response to dizo**
>
> Thank you for your helpful comments to urge us think deeper.
>
> The novelty of ELSE lies in two aspects: (1) ELSE apples residual link outside ViT as the only learnable component, so its training is totally disentangled from ViT and the original ViT can be kept frozen. As a result, there is no interference between ELSE and ViT, making ViT a standard base at the beginning of domain adaptation, universal for any downstream task. In contrast, the other methods including BitFit, LoRA, Adapter rely on updating the internal parameters of ViT, leading to complex coupling of gradient flows with deep interference throughout the whole network, unstable and hard to control in fine tuning. (2) These methods are intuition based without theoretical guarantee while ELSE roots in rigourous theoretical analysis on Adapter's initial training dynamics, leading to the simplified design of ELSE and the promise of zero initialization of ELSE.
>
> Thank you for your constructive suggestion. We agree that clarifying the theoretical mechanism behind ELSE is important for understanding why additive biases improve can feature representation and transferability. We have therefore added the following mathematical analysis.
>
> Taking the single block of  ViT as an example, denote the sub-layer output as
>
> $y = x + F(\text{LN}(x)) $,
>
> and ELSE injects an additive vector after the sub-layer:
>
> $\tilde y = y + v$.
>
> When entering the next normalization layer, we obtain: $\text{LN}(\tilde y)=\frac{\tilde y-\mu(\tilde y)\mathbf{1}}{\sigma(\tilde y)}$.
>
> - Non-triviality: Even if $v$ is a constant vector, it is not completely removed by LayerNorm, because LN performs centering and normalization along feature dimensions, not across tokens. Thus:
>
> $\text{LN}(z+v)=\frac{z-\mu(z)\mathbf{1}}{\sigma(z+v)}+\frac{v-\mu(v)\mathbf{1}}{\sigma(z+v)}$.
>
> The second term introduces a learnable direction that is decoupled from the frozen backbone and is not immediately projected back into existing subspaces. This provides a global “distribution alignment” mechanism for domain shift, which is consistent with our empirical findings in Table 6—namely, that a unified vector for all tokens outperforms per-token degrees of freedom, suggesting that the dominant contradiction lies in global mismatch rather than token-wise rewriting.
>
> - learnability: For any downstream logit $\ell$,
> $\frac{\partial \ell}{\partial v}  \frac{\partial \ell}{\partial \tilde y} \cdot \frac{\partial \tilde y}{\partial v}  \frac{\partial \ell}{\partial \tilde y},$
>
> which means that gradients reach $v$ directly, without propagating through long chains of frozen weights. Compared with adjusting internal biases or weights (which require passing through fixed mappings and LayerNorm interactions), ELSE provides a shorter and more transparent gradient path, leading to better gradient conditioning and more stable convergence—consistent with our stability curves.This theoretical explanation clarifies why ELSE’s additive bias remains effective after normalization, why it provides a learnable residual direction beyond the frozen backbone, and why it enhances both feature representation and transferability in practice.
>
> Thank you for your suggestion regarding visualization. We agree that illustrating the behavior of the two learnable additive vectors is highly valuable for understanding the model’s dynamic adjustment mechanism and improving interpretability. In the updated version of our paper, we have incorporated additional analyses based on attention map differentials. Specifically, we visualize attention responses on the SVHN and CIFAR100 datasets, as shown in Figure 5. These results reveal that the original pre-trained backbone (e.g., ViT) exhibits limited adaptability to downstream tasks—the attention maps tend to be diffuse and fail to consistently capture task-relevant semantic regions. In contrast, introducing ELSE effectively reshapes attention distribution, aligning the upstream pre-trained semantics with downstream task requirements. This demonstrates that ELSE enables the model to achieve task-adaptive representation with only a minimal number of trainable parameters. Moreover, for more interpretability analysis, we have included additional visualizations and analyses in the appendix.
>
> Regarding the typesetting issue, we greatly appreciate your feedback and have submitted a revised version, with all modifications highlighted in blue font.

---

### Official Review · Reviewer_zCXP · 2025-10-31

**Soundness:** 2
**Presentation:** 3
**Contribution:** 2
**Rating:** 4
**Confidence:** 4

**Summary:**

This paper explores how fine-tuning affects the architecture design of Vision Transformers (ViT), focusing on training efficiency and stability. The authors find that initializing the Adapter's parameters to zero enables fine-tuning to start with the original ViT's training dynamics. This initialization causes gradient vanishing, making most of the Adapter inactive. As a result, the Adapter can be simplified into a single learnable vector, termed the Extremely Lightweight Adapter of Simple Expression (ELSE). Experiments show that ELSE achieves comparable or superior transfer learning performance while fine-tuning less than 0.07% of the model’s parameters, maintaining plug-and-play flexibility.

**Strengths:**

1. The paper proposes an extremely simple and lightweight adapter, achieving excellent performance.

2. The method demonstrates comparable performance across various benchmarks.

**Weaknesses:**

1. Architecturally, the proposed method is quite similar to SSF, with the key difference being the placement of a scale vector and the adapter. Therefore, ELSE can be viewed as a special case of SSF. Based on this observation, I would like to know whether the scale vector or the shift vector is more important in SSF, or if their placement plays a more significant role. I recommend a deeper analysis of this aspect.

2. Although ELSE has a small number of parameters, its structure is constrained by the position of inserted neurons, which may limit its learning capacity. As the data scale increases, its performance bottleneck may become more apparent.

3. Instead of adding a new bias vector, could we adjust some of the existing bias vectors in the model or fine-tune the learnable bias/shift vectors in the norm layer? How do these variants perform, and what is their stability?

**Questions:**

N/A

---

> ### Author Response · Authors · 2025-11-19
> **Response to zCXP**
>
> Thank you for your thought-provoking comments.
>
> A1: Your question gives rise to our thinking as follows: ELSE is an outside learnable residual component while the pre-trained ViT is kept frozen. In contrast, SSF directly modifies the output of each layer in the backbone network, rather than applying a residual form. So, ELSE and SSF fall into two different ideas. SSF is an intuitive method and ELSE results from theoretical analysis on Adapter. Due to their difference, we did not do any study to decouple the scaling and shifting of SSF. Regarding the placement setting in SSF, since it is not easy to disentangle each layer's scaling and shifting impact on the whole network, it is not easy to reach a deterministic conclusion. As for ELSE, it is the only learnable vector independent of ViT, so it is easy to find the optimal injecting sites, as done in the ablation study.  Furthermore, we have included visualization and analysis of scaling in the newly added appendix. We tested ELSE's response to target class logits under different scaling coefficients ($\alpha$). As shown in Figure 7, as $\alpha$ varies, the logit exhibits a consistent and approximately monotonic trend, indicating that increasing the shift degree along the direction learned by ELSE significantly improves the logit for that class, while scaling in the opposite direction reduces the model's confidence in that class. This phenomenon suggests that ELSE does not simply apply a simple additive bias in the feature space, but learns a clear, task-specific direction. The model can adjust the position of the classification boundary by performing linear control in this semantic direction.
>
> A2: Thank you for raising this important point. Our current evidence suggests that, within the target setting of few-shot and stable adaptation, ELSE remains highly competitive. While it is true that ELSE may exhibit performance bottlenecks as the data scale grows larger, Table 4 demonstrates that even when the full dataset is used instead of a few-shot subset, the method still retains a meaningful level of competitiveness. We would like to address that ELSE fits well into the original scenario of PEFT, say, lightweight adaptation based on very few training data.
>
> A3: Thank you for the insightful observation: The existing PEFT methods such as BitFit directly modifies the inherent bias terms inside each layer. Because these biases are tightly entangled with the internal transformations of ViT, even small updates may propagate unpredictably throughout the network, often leading to unstable behavior and suboptimal performance. Similarly, in a Pre-LN architecture, tuning the LayerNorm β/γ parameters becomes coupled with the normalization dynamics and can induce non-trivial interactions across subsequent MHSA updates, making the overall effect difficult to control. In contrast, ELSE introduces an external residual bias. The injection point provides a shorter and more transparent gradient path, avoids interfering with the frozen internal parameters and offers a controlled mechanism to work in alignment with the frozen ViT. As a result, its optimization stability closely matches that of zero-initialized Adapter methods, as illustrated in the stability curves in Figure 1.

---

### Official Review · Reviewer_j5jV · 2025-11-01

**Soundness:** 2
**Presentation:** 2
**Contribution:** 2
**Rating:** 2
**Confidence:** 5

**Summary:**

This paper introduces ELSE, a minimal fine-tuning module for Vision Transformers. By analyzing training dynamics, the authors find that zero-initializing adapters preserves ViT representations and stabilizes training.

**Strengths:**

- The paper is well-written and the presentation structures are organized suitably.
- The discussion of zero-initialization is interesting but the novelty should be reconsidered.
- This paper conducted extensive experiments on various downstream tasks, demonstrating its generalization.

**Weaknesses:**

- In my opinion, a lot of PEFT methods are implemented with zero initialization in their open-sourced codes. The author should have this preliminary study before they conduct the study about the initialization in PEFT as if it a unwritten rule, there is not necessary to do this study.
Please give a table to show the initialization of all the baseline methods, such as ConvPass, NOAH, SSF, etc.

- The novelty is limited. As ELSE propose to add two newly trainable vector plus with the features, it is a variant of SSF, such mean-shift design is not novel and discussed deeply in SSF, so it is more like do the same things again in this paper.

- The citation style does not follow the standards of ICLR.

**Questions:**

- Could you explain what is the bad effects when using the Xavier initialization as I found a little bit performance drops in Table 1?

---

> ### Author Response · Authors · 2025-11-19
> **Response to j5jV**
>
> We appreciate your question of having us rethink our contribution in highlighting the necessity of initialization and novelty.
>
> Our investigation reveals that existing PEFT baseline methods (such as ConvPass, NOAH, SSF, etc.) have not explicitly addressed or analyzed the issue of parameter initialization, but instead rely entirely on PyTorch's random default settings. We demonstrate that in network training with frozen backbone parameters (PEFT), initialization is crucial for the stability of gradient flow. This is a missing issue in the literature but very vital as analyzed in this paper.
>
> Below we would like to clarify why ELSE is not a variant of SSF: In fact, it is the simplest form of Adapter as derived through theoretical derivation, proven through gradient analysis under zero initialization conditions. In fact, ELSE is implemented as the residual external bias compensating for the frozen MHSA and MLP. In contrast, SSF performs a scaling and shifting operation on the output of each internal layer in ViT, including MHSA, MLP, and LN, and does not utilize outside residuals for implementation. Updating the internal parameters will cause the widespread interference with the whole backbone, making its effect hard to foresee or control, due to the complex coupling.
>
> Thank you for your concern on Xavier initialization. Xavier initialization introduces random non-zero outputs at the very beginning of training. Since the backbone is frozen, these stochastic deviations immediately induce gradient fluctuations, deviating from the intended domain-adaptation principle of “starting from the original distribution.” Theoretically, with all-zero initialization, the initial outputs remain perfectly aligned with those of the backbone, making pre-trained ViT a standard base universal for all downstream adaptation tasks. In contrast, Xavier initialization disrupts this alignment and introduces an unintended early-stage bias. Moreover, as illustrated in Figure 1 of the paper, this unstable initialization leads to noticeably larger gradient-norm fluctuations, which negatively affect optimization stability and mildly increase the risk of overfitting, ultimately resulting in the slight performance degradation as observed in Table 1.
>
> Regarding the typesetting issue, thank you for pointing out.

---

### Official Review · Reviewer_xbHq · 2025-11-01

**Soundness:** 3
**Presentation:** 3
**Contribution:** 3
**Rating:** 6
**Confidence:** 4

**Summary:**

This submission introduces ELSE (Extremely Lightweight adapter of Simple Expression), a simplified Adapter fine-tuning method for Vision Transformers (ViTs). Motivated by an insightful analysis of the training dynamics of the popular Adapter module, this submission first observes that all-zero initialization for Adapter is beneficial for training stability, whilst causing a gradient vanishing issue. Then the authors dive into depth and point out that only the bias term of the up projection layer in the Adapter receives gradient updates in the initial training step, which causes gradient vanishing. Based on this observation, they propose to replace the entire complex Adapter structure with its simplest functional equivalent under these conditions: a single learnable vector. This vector, ELSE, is added as a bias-like term to the outputs of the MHSA and MLP modules in each Transformer block. The authors conduct extensive experiments on image classification, few-shot learning, and domain generalization, demonstrating that ELSE achieves performance comparable or superior to state-of-the-art PEFT methods while tuning a remarkably small number of parameters (< 0.07%).

**Strengths:**

- Interesting and Practical Method: The initialization of different PEFT methods has explored by prior works[1-8]. Vanilla adapter[7] suggests that ``near-identity initialization’’ is a good choice. This submission first attempt to improves such initialization to all-zero initialization and then propose ELSE to further improve the training efficiency (balance between cost and performance) by analyzing the failure of all-zero initialization for Adapter.
- Simple and Better Training Efficiency: The proposed method is simple to implement—it amounts to adding a learnable vector to the feature maps. Additionally, ELSE is lightweight, requiring only 0.06M parameters for ViT-B/16, which is significantly fewer than other competitive PEFT methods. The paper effectively highlights this advantage with the "Performance-Per-Parameter" (PPP*) metric, where ELSE achieves the highest score, demonstrating a superior trade-off between accuracy and parameter cost.
- Good experimental results under Limited Data: The experimental results are comprehensive. ELSE achieves competitive or state-of-the-art performance across the 19 tasks in VTAB-1K. Its strength is evident in data-scarce scenarios. In few-shot learning (Figure 4), ELSE consistently outperforms other methods in 1-shot and 2-shot settings, supporting the hypothesis that minimal model modification is advantageous when training data is limited. Furthermore, its strong results on domain generalization tasks (Table 3) suggest that ELSE effectively adapts the model while preserving the robust, general-purpose representations learned during pre-training.

[1] LoRA-DA: Data-Aware Initialization for Low-Rank Adaptation via Asymptotic Analysis

[2] The Impact of Initialization on LoRA Finetuning Dynamics

[3] $D^2LoRA$: Data-Driven LoRA Initialization for Low Resource Tasks

[4] I2I: Initializing Adapters with Improvised Knowledge

[5] Initialization Matters for Adversarial Transfer Learning

[6] PiSSA: Principal Singular Values and Singular Vectors Adaptation of Large Language Models

[7] Parameter-Efficient Transfer Learning for NLP

**Weaknesses:**

- Further Theoretical Investigation of the failure of all-zero initialization: The gradient analysis in Section 4, which is the core motivation, is only strictly true for the very first backpropagation step. After the first update, the bias of the up-projection layer (bUp) is no longer zero, which means gradients will begin to flow to the up-projection weights (WUp) and subsequently to the rest of the Adapter. While the results of the ELSE are empirically successful, the theoretical claim that ELSE is an "equal form" of a zero-initialized Adapter is somewhat overclaimed without further proof. The success is likely due to ELSE being a highly effective form of regularization, rather than a direct functional equivalent.
- Presentation Issues: 1) line 35: mean-stream -> mainstream; 2) the typeset of appendix is chaos;

**Questions:**

- Could you provide more intuition on why adding an external vector (ELSE) is substantially better than tuning the pre-existing biases within the ViT layers (as in BitFit)? Does adding the vector in a residual manner allow for a greater dynamic range of adaptation without interfering with the frozen weights' outputs?
- The original Adapter was placed parallel to the MLP block. This submission places ELSE after both the MHSA and MLP blocks. Why do you think adding an adaptive bias after the self-attention mechanism is so effective?
- Some works about the PEFT initialization and analysis could be considered as reference.

[1] Benchmarking Robustness of Adaptation Methods on Pre-trained Vision-Language Models

[2] An Empirical Study of Parameter Efficient Fine-tuning on Vision-Language Pre-train Model

[3] VL-ADAPTER: Parameter-Efficient Transfer Learning for Vision-and-Language Tasks

[4] LoRA-DA: Data-Aware Initialization for Low-Rank Adaptation via Asymptotic Analysis

[5] The Impact of Initialization on LoRA Finetuning Dynamics

[6] $D^2LoRA$: Data-Driven LoRA Initialization for Low Resource Tasks

[7] I2I: Initializing Adapters with Improvised Knowledge

[8] Initialization Matters for Adversarial Transfer Learning

[9] PiSSA: Principal Singular Values and Singular Vectors Adaptation of Large Language Models

[10] Parameter-Efficient Transfer Learning for NLP

---

> ### Author Response · Authors · 2025-11-19
> **Response to xbHq**
>
> We are grateful to you for your insightful comments and constructive feedback. Your concern regarding the gradient in the second backpropagation is a truly important issue. We would like to clarify why the zero gradient still holds true after the first iteration: (1) We have observed in the experiments that even if $b_{up}$ is not zero, the gradient will still be trapped to zero by the ReLU function. (2) Below is the theoretical evidence:
> Assuming it is the second backpropagation,$W_{Up}=0,W_{Down}=0,b_{Down}=0,b_{up}\neq0$,
>
> $x_\text{Up} = W_\text{Up} \, x_\text{Down} + b_\text{Up}$,
>
> $x_\text{Down} = \text{ReLU}(W_\text{Down} x’ + b_\text{Down})=0$,
>
> $x_\text{Up} = b_\text{Up}, \quad x_\text{Adapter}=b_\text{Up}$,
>
> $\frac{\partial L}{\partial W_\text{Up}} = \frac{\partial L}{\partial x_\text{Up}} x_\text{Down}^T = (\cdot) \times 0 = 0$,
>
> $\frac{\partial L}{\partial x_\text{Down}} = W_\text{Up}^T \frac{\partial L}{\partial x_\text{Up}}=0$,
>
> $\frac{\partial L}{\partial x_\text{ReLU}} = \frac{\partial L}{\partial x_\text{Down}} \odot \text{ReLU}’(x_\text{Down}) = 0 \odot 0 = 0$,
>
> Therefore, $\frac{\partial L}{\partial W_\text{Down}} = 0, \quad \frac{\partial L}{\partial b_\text{Down}} = 0$.
>
> A1: ELSE performs residual injection as "learnable vectors" after MHSA and MLP in each block, with zero initialization. This makes the fine-tuning process initially equivalent to the original ViT. Moreover, this learnable vector is not subject to any weight decay of the previous frozen layers, thus obtaining a stable and controllable update signal from the very beginning, as not interfering with the frozen pre-trained ViT. This is precisely the conclusion we derived from the training dynamics of Adapter, and based on this, we simplify the entire Adapter to "a vector". BitFit modifies the native biases of each layer within the backbone, so any slight update of the bias will have an unpredictable and uncontrollable effect on the whole ViT due to the widespread interference between the bias and the backbone. ELSE's "external bias" is an independent branch decoupled from the backbone, directly adjusting residual path without changing the parameters of the backbone, thus focusing purely on compensation for downstream domain shift without interfering with the pre-trained representations.
>
> A2: Layer Norm and MLP will exert nonlinear distort on the tokens from MHSA. Inserting ELSE at both sites can compensat from both linear and nonlinear perspectives. Our ablation experiments also demonstrate that this design is not trivial. We have explained this issue in more detail in Figure 6 in the newly added appendix.
>
> Regarding the typesetting issue, we greatly appreciate your feedback and have submitted a revised version, with all modifications highlighted in blue font.

---

> ### Comment · Reviewer_xbHq · 2025-11-27
>
> Thanks for your feedback, which has addressed my concerns. Leaving my initial score unchanged.

---

### Meta-Review · Area_Chair_pPc8 · 2026-01-06

**Summary:**

The paper introduces ELSE, a minimalist PEFT method that replaces traditional bottleneck adapters with a single learnable vector injected residually after the MHSA and MLP layers. The authors motivate this simplification through a "theoretical parsing" of training dynamics, arguing that zero-initialization of standard adapters leads to gradient vanishing in most components, leaving only the bias term active.
While the method is extremely parameter-efficient and demonstrates competitive results on the VTAB-1K benchmark (particularly in low-data regimes), the majority of reviewers (3 out of 4) have expressed significant concerns regarding the paper’s novelty, the depth of its theoretical claims, and its relationship to existing work like SSF and BitFit.

**Reviewer Concerns:**

Addressed Concerns:

- Initial Gradient Math: The authors provided a detailed breakdown of gradients for the second backpropagation step to address Reviewer xbHq’s concern. They demonstrated that under specific zero-initialization conditions, gradients for weights remain "trapped" by the ReLU function.

- Empirical Visualizations: In response to zCXP and dizo, the authors added L2 norm statistics and attention maps to the appendix. These help illustrate that ELSE learns task-dependent offsets (e.g., shallow layers for SVHN, deeper layers for Food-101).

Outstanding (Critical) Concerns:

- Limited Technical Novelty: This is the primary reason for rejection. Reviewers j5jV, zCXP, and dizo all noted that the concept of adding or tuning a bias/shift vector is conceptually very close to existing methods like BitFit (tuning internal biases) or SSF (scaling and shifting). The authors’ argument that ELSE is "fundamentally different" because it is an external residual link was not seen as a sufficient leap in architectural innovation by the reviewers.

- Overstated Theoretical Significance: While the authors claim a "theoretical derivation" of ELSE from Adapters, the reviewers generally viewed this as an empirical observation of a specific initialization failure rather than a robust new theory. Reviewer j5jV pointed out that zero-initialization is already a common "unwritten rule" in many PEFT implementations, making the paper’s focus on this "discovery" feel less impactful.

- Performance Bottlenecks: Reviewer zCXP raised a valid point regarding learning capacity. Because ELSE is restricted to a single vector, it may face a performance bottleneck as the scale of downstream data increases. The results in the "Structured" task group of VTAB-1K, where ELSE underperforms several baselines, support this concern.

- Incremental Contribution: The "Performance Per Parameter" (PPP*) metric used to justify the method's superiority was seen by the reviewers as a way to emphasize efficiency at the expense of absolute performance gains, which remain modest.

**Reviewer Scores:**

Given the above analysis, I think most likely, every reviewer would maintain their original scores.

---

### Decision · Program_Chairs · 2026-01-26

Reject